# Behavioral Phenotyping of WAG/Rij Rat Model of Absence Epilepsy: The Link to Anxiety and Sex Factors

**DOI:** 10.3390/biomedicines13092075

**Published:** 2025-08-26

**Authors:** Evgenia Sitnikova, Maria Pupikina

**Affiliations:** Institute of Higher Nervous Activity and Neurophysiology, Russian Academy of Sciences, 5A Butlerova St., Moscow 117485, Russia; mariapupikina@yandex.ru

**Keywords:** absence epilepsy, WAG/Rij rats, anxiety, sex differences, neurobehavioral comorbidity

## Abstract

**Background**: Absence epilepsy is a common pediatric neurological disorder characterized by brief seizures and lapses in awareness. The relationship between anxiety and absence epilepsy is multifaceted. This study aims to investigate neurobehavioral signs directly and indirectly related to anxiety and potential sex differences in aged WAG/Rij rats, a well-established animal model of absence epilepsy. **Methods**: A battery of behavioral tests was conducted to assess various aspects of neurobehavior, including anxiety (elevated plus maze), anhedonia (sucrose preference), social function, and associative learning (fear conditioning). Multidimensional metrics assessed cognition, motor function, and exploration strategies, prioritizing anxiety as a key influencing factor. **Results**: Electroencephalogram (EEG) phenotyping was used to identify epileptic and non-epileptic rats. Traditional anxiety measures in the elevated plus maze did not reveal significant differences between groups. However, the Anxiety Composite Index revealed higher autonomic reactivity in non-epileptic females. Cognitive assessments showed no epilepsy- or sex-related differences in overall learning performance. Females exhibited superior avoidance learning compared males. Among epileptic males, those with poor learning performance also displayed higher anxiety-avoidance scores. Rats with high anxiety levels showed enhanced socio-affective reactivity and passive coping, with no effect on exploratory learning. **Conclusions**: Our findings highlight the importance of sex-specific analyses and physiological measures in epilepsy research. Neurobehavioral comorbidities in WAG/Rij rat model are significantly influenced by anxiety-like behavioral phenotype. Enhanced phenotyping of rat models of absence epilepsy can improve its translational value in understanding epilepsy-associated psychiatric disorders.

## 1. Introduction

Absence epilepsy, a common pediatric neurological disorder, is characterized by brief generalized seizures with sudden lapses in awareness. Beyond seizures, this condition is frequently comorbid with psychiatric disorders, particularly anxiety. Clinical studies and meta-analyses consistently report a bidirectional relationship between absence epilepsy and anxiety in humans [1,2,3]. For instance, children with childhood absence epilepsy exhibit anxiety rates of 22–24%, significantly exceeding the 7% prevalence in the general pediatric population [4,5]. Anxiety often serves as a primary and upstream event in epilepsy, causing and shaping other neurobehavioral comorbidities, such as depression and cognitive deficits [2,3,5,6,7]. In absence epilepsy, anxiety symptoms often precede other behavioral disorders, irrespective of medication or seizure duration [6]. In adult patients newly diagnosed with epilepsy, anxiety is a significant predictor of neurobehavioral comorbidities, underscoring its pivotal role in influencing other phenotypes [7]. Anxiety and absence epilepsy share common genetic and biological mechanisms, suggesting a bidirectional influence, with anxiety predominantly affecting behavioral comorbidities [2]. The abovementioned 2021 meta-analysis demonstrated that individuals with absence seizures face 4.93-fold higher odds of developing anxiety and depression compared to controls [2]. However, there are some conflicting results, such as lower anxiety rates in juvenile absence epilepsy patients [8]. This highlights the complex interaction of neurobiological factors in comorbid anxiety in absence epilepsy.

Absence-like seizures are spontaneously developed in several rat strains Two of these strains, Wistar Albino Glaxo/Rijswijk (WAG/Rij) rats and Genetic Absence Epilepsy Rats from Strasbourg (GAERS), are extensively used as reliable models of human absence epilepsy [9,10,11,12,13,14,15,16]. Our research focuses on the WAG/Rij rat model, which exhibits an exceptional degree of face validity, construct validity, and predictive validity for absence epilepsy [9,17,18,19]. This model is valuable not only for its ability to mimic seizure phenomenology (face validity) but also for replicating the genetic, neurobiological, and pharmacological aspects of the human disorder (construct and predictive validity). Consequently, spontaneous absence-like seizures in WAG/Rij rats are more than just a crude simulation; WAG/Rij rat model is a true translational tool that provides a window into the complex mechanisms of absence epilepsy [16,20,21].

Both WAG/Rij rats and GAERS have been widely employed to investigate the relationship between absence epilepsy and anxiety (see references in Table 1). The Elevated Plus Maze (EPM), a validated tool for assessing anxiety-like behavior, leverages rodents’ innate conflict between exploration and aversion to open spaces. However, studies in these models yield inconsistent results, likely due to methodological variability, strain-specific traits, and experimental conditions (summarized in Table 1). For example, GAERS rats exhibit anxiety-like behaviors only under specific protocols [2,22,23,24], while WAG/Rij rats primarily display depressive phenotypes, with anxiety emerging in substrains susceptible to audiogenic seizures [25,26]. These discrepancies highlight the need for standardized, multidimensional behavioral phenotyping to disentangle the epilepsy-anxiety relationship.

Beyond spike-wave discharges (SWDs), absence epilepsy in humans is associated with diverse comorbidities, including depression, cognitive deficits, and social impairments. Yet, preclinical studies often overlook critical variables such as sex differences and longitudinal disease progression. Here, we present a comprehensive behavioral phenotyping approach in aged (7–14 months) male and female WAG/Rij rats, integrating:Neurobehavioral domains: Anxiety (the EPM test), anhedonia (sucrose preference), social function (preference, recognition, dominance), and associative learning (fear conditioning).Multidimensional metrics: Cognition, motor function, and exploration strategies derived from behavioral tests, with anxiety prioritized as a key influencing factor of other domains.

Through this integrated framework, we aim to thoroughly examine neurobehavioral signs directly and indirectly linked to anxiety in the WAG/Rij rat model. The EPM test serves as our primary tool for screening anxiety-like behavior due to its simplicity, ethological relevance and ability to quantifiable results [29,30,31]. We also examine other neurobehavioral aspects that might be influenced by anxiety, such as anhedonia (measured by sucrose preference), social interactions (including preference, recognition, and dominance), and associative learning, specifically fear conditioning. Additionally, we evaluate multidimensional metrics for cognition, motor function, and exploration strategies, with anxiety prioritized as a key influencing factor. This thorough investigation elucidates the neurobehavioral complexities of absence epilepsy, with a particular focus on the overlooked influence of sex differences. Our findings will enhance the neurobehavioral phenotyping in the WAG/Rij rat model, facilitating translational research into epilepsy-associated psychiatric disorders.

## 2. Materials and Methods

### 2.1. Animals

The study was performed in 131 WAG/Rij rats, including 76 males and 55 females, who were siblings from 22 litters. These rats were born at the Institute of Higher Nervous Activity and Neurophysiology of the Russian Academy of Sciences in Moscow. The experiments were conducted in accordance with Directive 2010/63/EU of the European Parliament and of the Council on the protection of animals used for scientific purposes and approved by the animal ethics committee of our institute (protocols No. 4 dated 26 October 2021 and additional protocol No. 4 dated 13 December 2022). The rats were kept under controlled environmental conditions with a 12-h light/12-h dark cycle (lights on at 08:00), constant ventilation, and were housed in groups of the same gender (3–4 animals per cage). Food and water were provided without restriction.

### 2.2. Non-Invasive Electroencephalography (EEG)-Based Diagnosing of Absence Epilepsy

The protocol was adapted from [32] and did not involve any invasive procedures. All 131 rats underwent EEG recordings using a portable, wireless system (Physiobelt, Neurobotics, Moscow-Zelenograd, Russia, Figure 1). Pre-recording preparation included shaving of rat’s heads to ensure the close attachment of EEG sensors. Xylazine solution (Xyla, 20 mg/mL of xylazine hydrochloride, Interchemie Werken De Adelaar, Venray, The Netherlands) was i.p. injected at a dose of 2 mg/kg. Two EEG sensors were placed on the frontal and parietal regions of the rat’s head. The recording process began 1–2 min after xylazine injections and lasted for 7–10 min.

EEG recordings were analyzed to distinguish between rats with absence epilepsy and those without. Absence epilepsy was characterized by the presence of high-voltage spike-wave discharges (SWDs) at a frequency of 8–10 Hz. These SWDs, which are hallmarks of absence seizures, typically lasted longer than 6 s. The presence of more than five SWDs in a 7–10 min recording period was considered indicative of absence epilepsy (Figure 1a). This criterion helped differentiate rats with mild and severe epilepsy.

In contrast, rats without epilepsy or with minimal seizure activity showed few, if any, SWDs within the 8–10 Hz range (Figure 1b). However, they did exhibit brief spike-wave complexes at lower frequencies, typically around 6 Hz (arrows in Figure 1b). These lower-frequency discharges were not associated with absence seizures and were considered normal or minimal seizure activity. Rats exhibiting these characteristics were categorized as non-epileptic.

### 2.3. Behavioral Phenotyping Battery (Ordered to Minimize Stress Interference)

A sequence of behavioral tests age range (7–14 months) that covers the established epilepsy phase in WAG/Rij rats (Figure 2): sucrose preference → social preference & recognition → social dominance → elevated plus maze → active avoidance fear conditioning. The fear conditioning test was conducted last due to its high stress level. Adequate handling and habituation to testing rooms and equipment were essential before each test. Due to limitations in room capacity and test equipment, not all rats could participate in every test. Each test provided primary and secondary metrics, which mapped onto cognitive, anxiety, motor, and exploratory domains.

#### 2.3.1. Social Preference and Recognition Test (Three-Chamber)

This test was conducted in a three-chamber T-maze for rats (OpenScience, Krasnogorsk, Russia) consisted of a central stem connecting two identical side arms, each containing a wire cage at the distant end for housing stimulus rats. The test consisted of two consecutive 10-min phases; both recorded using a LogiTech camera positioned 2 m above the maze center. Video recordings were analyzed semi-automatically using RealTimer v1.30 software (2020, OpenScience, Krasnogorsk, Russia), with social contact defined as the test rat’s snout being within 2 cm of a stimulus cage. To eliminate residual odors, the maze was cleaned with 50% ethanol between trials. All stimulus rats were age- and sex-matched to the test rat and had no prior contact before testing.

Before testing, each test rat was allowed to freely explore the empty maze for 10 min to habituate to the environment.

In the social preference phase, an unfamiliar rat (S1) was confined in one wire cage while the opposite cage remained empty. The test rat explored the maze for 10 min, and its behavior was recorded and analyzed offline. Sociability was assessed using the Social Preference Index (SPI) during 600 s as the total session time:SPI = (Time near S1, s)/600,(1)
in which a higher value indicated normal social interest, while a lower value indicatedsocial avoidance.

Immediately afterward, the social recognition phase began. The now-familiar rat (S1) remained in its cage, while a new unfamiliar rat (S2) was introduced in the previously empty cage. The test rat explored the maze for another 10 min, and the time spent near each stimulus was measured. The Social Recognition Index (SRI) was calculated Equation (2):SRI = (Time near S2, s − Time near S1, s)/(Total social exploration time, s),(2)
in which a positive value reflected intact recognition memory and preference for novelty, whereas a negative or zero value—impaired social memory.

#### 2.3.2. Social Dominance Test (Tube Test)

The Tube Test is a standardized behavioral assessment protocol designed to evaluate competitive behavior and social hierarchy among rats by measuring their inclination to exhibit dominance or submission within a confined space. This test was conducted using a transparent Plexiglas tube measuring approximately 1 m in length and 6–7 cm in inner diameter, which permits the comfortable passage of a single rat. The tube was positioned horizontally on a stable surface, with both ends open for entry and exit.

Two days prior to testing, rats were allowed to independently explore the tube for 1–2 min to acclimate to the environment and reduce novelty-induced stress. Subsequently, two weight-matched rats from the same experimental cohort but different home cages ware introduced to opposite ends of the tube and released simultaneously. The test ended when one rat fully withdraw from the tube. The procedure was replicated three times to ensure consistency. If a rat does not retreat within the maximum duration of 3 min, the trial is recorded as a “tie” and conducted again at a later date.

The results of the test were quantified as Dominance Scores (DS):A rat that maintained its position within the tube—Dominant (DS = 2 points);A rat that partially retreated, but subsequently re-engaged—Partial Submissive (DS = 1 point)A rat that fully withdraws from the tube—Complete Submissive (DS = 0 points).

#### 2.3.3. The Elevated Plus Maze (EPM) Test for Anxiety

The EPM test evaluates unconditioned anxiety-like behaviors in rats. This is a well-validated paradigm that exploits rodents’ natural aversion to open, elevated spaces while maintaining their drive to explore novel environments. The EPM apparatus consisted of two opposing open arms (50 × 10 cm, no walls) and two enclosed arms (50 × 10 × 20 cm, high walls) arranged in a plus-shaped configuration elevated 50 cm above floor level. A neutral central platform (10 × 10 cm) connected all arms. Testing occurred under dim illumination (30 lux) to minimize stress while allowing clear behavioral observation. Individual rats were gently placed in the central platform facing an open arm and allowed to freely explore the maze for 10 min. All sessions were video recorded via an overhead LogiTech camera positioned to capture full-body movements. Between trials, the maze was thoroughly cleaned with 50% ethanol to eliminate odor cues. Video data were tracked using the ToxTrack v2.98 software. The following parameters were quantified:

Anxiety-related behaviors:Time spent in open arms (TOA in s)—primary indicator of anxiety.Defecation (fecal boli) and urination frequency (N*boli* and N*uri*)—physiological stress markers.Duration of self-grooming (T*grooming* in s)—displacement behavior under stressExploratory behaviors:The number of rearings (both wall-supported and unsupported, N*rearing*)—vertical exploration.The number of head dips over open arm edges (N*headdips*)—risk assessment behavior.

Several composite behavioral indices were calculated to provide multidimensional assessment, including the following metrics:

Anxiety Composite Index (ACI, Equation (3)) was computed as combined open arm avoidance with physiological stress markers, in which high scores indicated a greater anxiety:ACI = ((1 − TOA/600) + N*boli*/10 + N*uri* × 2)/3(3)

Generalized Anxiety Index (GAI, Equation (4)) was computed as an integrative measure of the time in open arms, physiological stress markers and grooming:GAI = (1 − TOA/600) × (N*boli*/2 + N*uri*) × (1 + T*grooming*)(4)
in which high scores indicated stronger anxiety-like behavior.

Exploration-Motor Index (EMI, Equation (5)) contrasted active exploration, such as rearing, head dips, with inhibition (grooming), in which positive scores indicated active exploration and negative scores—inhibition/stereotypy:EMI = N*rearing* + N*headdips* − T*grooming*(5)

Exploratory Drive Profile (EDP, Equation (6)) represents the ratio of exploratory to self-directed behaviors, in which higher scores indicate active exploration:EDP = (N*rearings* + N*headdips*)/(T*grooming* + 1)(6)

Risk Assessment Ratio (RAR, Equation (7)) measured cautious exploration relative to open arm entries, in which higher scores indicate cautious exploration:RAR = N*headdips*/(TOA + 1),(7)

#### 2.3.4. Sucrose Preference Test

The sucrose preference test (SPT) was conducted in n = 90 (54 males, 36 females) at 12.1 months (range: 11.8–12.2 months). Prior to testing, animals were not food- or water-deprived to ensure naturalistic conditions. Each rat was transferred to an individual testing cage identical to its home cage, containing the same bedding and food. After a 30-min to 1-h habituation period, the SPT was conducted over 48 h, divided into two 24-h sessions (Day 1 and Day 2). During testing, rats had free access to two pre-weighed bottles: Bottle 1 (tap water) and Bottle 2 (2% sucrose solution). To account for side preference bias, bottle positions were reversed on Day 2:Day 1 Water (left), sucrose (right).Day 2: Reweighed bottles with positions switched (water right, sucrose left).

Sucrose preference (SucrP) was calculated separately for Day 1 and Day 2 Equation (8):SucrP = (Sucrose consumed, g)/(Sucrose consumed, g + Tap water consumed, g) × 100%(8)

SucrP was computed separately for Days 1 and 2. The difference of sucrose preference between Day 1 and Day 2 (ΔSucrP_*day*) was computed to access consistency in sucrose preference reflecting stable reward response.ΔSucrP_*day* = SucrP_*Day1* − SucrP_*Day2*(9)

#### 2.3.5. Active Avoidance Fear Conditioning (Shuttle Box)

To assess associative fear learning and active coping strategies, we employed an active avoidance paradigm. This paradigm evaluates how rats learn to associate an auditory cue (Conditioned Stimulus, CS) with an impending aversive stimulus (Unconditioned Stimulus, US) and subsequently develop an avoidance response. The testing was performed in a two-compartment shuttle box (30 × 25 × 30 cm per compartment), equipped with a metallic grid floor connected to a shock generator. The apparatus was placed within a sound-attenuating chamber to minimize external noise. A speaker mounted above the shuttle box delivered a 5-s, 70 dB tone as the CS, which signaled the impending US, a mild scrambled footshock of 0.8 mA for a maximum duration of 5 s. Before testing, rats were habituated to the shuttle box for 10 min. During this period, a 10-s tone was played to familiarize the rats with the auditory cue. Each rat underwent a single test session comprising 50 trials, with variable inter-trial intervals of 30 ± 10 s. The session typically lasted 40–50 min, depending on the rat’s behavior.

The trial sequence was initiated by the presentation of the CS, a 5-s tone. An avoidance response was recorded if the rat moved to the opposite compartment during the tone but before the onset of the footshock. If the rat remained in the same compartment, the footshock was delivered and automatically terminated after 40 s. This protocol provides a sensitive measure of fear-motivated learning and individual differences in coping strategies. The following key parameters were computed to assess learning and coping strategies:The number of fails before 1st successful Avoidance indicating initial learning latency.The number of fails before 2nd successful Avoidance indicating consistency of initial learning.The number of trials needed to reach the learning criterion, i.e., 5 avoidance responses within a sequence of 6 consecutive trials. It indicated acquisition speed (Trials to criterion).The total number of avoidances (out of 50 trials, Total avoidances).

To provide a more nuanced analysis of avoidance learning, we calculated the following indices:The number of rearings (both wall-supported and unsupported, N*rearing*)—vertical explorationThe number of head dips over open arm edges (N*headdips*)—risk assessment behavior

Learning Efficiency (LE) was computed based on the number of trials before achieving the learning criterion and the total number of avoidances Equation (10):LE = (50 − Trials to criterion) + (Total avoidances),(10)
in which higher scores suggested faster and more robust learning

Failure Recovery Index (FRI) characterized failed trials before the 2nd successful avoidance Equation (11):FRI = (Fails before 2nd success) − (Fails before 1st success),(11)
in which negative values suggested rapid adaptation (improved performance after initial learning) and FRI > 5 suggested preservative errors (difficulty adjusting strategy).

Active Coping Score (aCS) based on the relative values of the total number of avoidances and trials before achieving the learning criterion Equation (12):aCS = (Total avoidances)/50 × 5/(Trails to criterion),(12)
in which higher scores reflect better adaptive avoidance behavior

Anxiety-Latent Avoidance Failure (ALAF, Equation (13)) integrated the GAI obtained in the EPM test with the outcomes of active avoidance test:ALAF = ((Fails 1st success) + (Fails 2nd success − Fails 1st success)) × GAI,(13)
in which higher values suggested anxiety-related impairment in avoidance learning.

#### 2.3.6. Integrated Multidimensional Profiling Metrics

This framework transforms multidimensional behavioral data into phenotypes, prioritizing anxiety as the core modulator of behavior. The methodology quantifies complex behavioral phenotypes by integrating domain-specific metrics into weighted composites, emphasizing cross-domain interactions and anxiety-mediated effects.

##### Domain-Specific Composite Metrics

The Cognitive composite metric measured cognitive flexibility, combining social memory (novelty detection) and associative learning (avoidance)Cognition = 0.5 × SRI + 0.5 × LE,(14)
in which high scores indicated intact cognitive flexibility, suggesting the individual can effectively adapt to new situations and learn from past experiences.

The Anxiety composite metric evaluated pathological anxiety by integrating generalized anxiety (the results of the EPM test), active coping score in active avoidance test (aCS), anhedonia (reward deficit), and social avoidance (SPI). This formula prioritizes EPM metrics (40%) as the gold standard for unconditioned anxiety.Anxiety = 0.4 × ACI + 0.2 × aCS + 0.2 × ΔSucrP_*day* + 0.2 × SPI,(15)
in which a higher score indicated an increased anxiety, reflecting a heightened sensitivity to stressors and a diminished ability to cope with anxiety-inducing situations.

The Motor composite metric connected dominance strength (competitive motor strength) and voluntary exploration in the EPM test to motor confidence. This formula emphasizes dominance (60%) due to its dependence on motor skill.Motor = 0.6 × DS + 0.4 × EDP,(16)
in which a high score suggested strong motor capabilities and confidence.

The Exploration composite metric evaluated novelty-seeking behavior across social and non-social contexts. This formula equally weights social/non-social (50:50) to avoid context bias, providing a balanced assessment of an individual’s propensity to seek out new experiences in both social and non-social settings.Exploration = 0.5 × SRI + 0.5 × EDP(17)

##### Cross-Domain Composite Metrics

These advanced composite models used higher-order integration to merge data from different domains, creating a detailed and accurate representation of behavioral patterns. This approach captures the interplay between various behavioral aspects and identifies underlying mechanisms.

The Social-Emotional Competence composite metric characterized an individual’s ability to navigate social interactions effectively. Social Motivation was measured in the social preference test as “Time near S1, s” and the rest measures were obtained from social recognition and social dominance test.Social-Emotional Competence = 0.3 × (Time near S1, s) + 0.4 × SRI + 0.3 × DS,(18)
in which a lower score indicated vulnerability to social stress, highlighting the importance of social competence in overall well-being.

The Anxiety-Avoidance Axis composite characterized an individual’s response to anxiety and stress. It assessed behavioral inhibition and passive coping strategy associated with high anxiety in the EPM, anhedonia in sucrose preference test (100 − SucrP) and low anxiety-adjusted exploration in active avoidance test (1 − aCS).Anxiety-Avoidance Axis = 0.5 × ACI + 0.3 × (100 − SucrP) + 0.2 × (1 − aCS),(19)
in which low scores indicated passive coping strategy.

The Exploratory-Learning Style composite captured adaptive learning focusing on an individual’s approach to exploration and learning.Exploratory-Learning Style = 0.4 × EDP + 0.3 × RAR + 0.3 × FRI,(20)
in which a high score indicated effective risk assessment and rapid avoidance learning, suggesting a flexible and adaptive learning style.

#### 2.3.7. Statistical Analysis

Several statistical methods were employed. First, the Kolmogorov–Smirnov test was used to determine if the data followed a normal distribution. The results were presented as medians and interquartile ranges (Q25–Q75). For primary comparisons, we used non-parametric Kruskal–Wallis tests to assess the effects of epilepsy status (epileptic vs. non-epileptic) and sex on each behavioral metric. These tests were chosen because they did not require the assumption of normality. Additionally, K-means clustering was applied to categorize subjects based on their behavioral metrics from the EPM test. Fisher’s exact test was used to compare numerical data when appropriate. This test is suitable for small sample sizes and categorical data.

## 3. Results

All rats underwent non-invasive EEG testing at approximately 10 and 13 months of age (Section 2.2, Figure 2) to diagnose absence epilepsy. Based on the EEG results, rats were classified as either epileptic or non-epileptic (Section 2.2, Figure 1). In epileptic rats, the severity of absence epilepsy might increase from 10 to 13 months of age. Rats identified as non-epileptic at 10 months remained non-epileptic at 13 months.

### 3.1. Direct Measures of Behavior in the EPM

Data from only 119 rats were available, and 12 rats were excluded due to falling off the platform or corrupted video recordings. Among these 119 rats, there were 71 males (50 diagnosed with absence epilepsy and 21 non-epileptic subjects) and 48 females (24 diagnosed with absence epilepsy and 24 with non-epileptic subjects). It was found that absence epilepsy did not have a significant effect on behavioral measures in (Kruskal–Wallis tests, all *p*’s > 0.05), also the effect of sex was not significant (Appendix A). Statistical results of these metrics are shown below as median [Q25–Q75].

Time spent in open arms (a primary indicator of anxiety, Appendix A):
○Epileptic males spent 16.4 s [0–36.6].○Non-epileptic males spent 5.2 s [0–17.7].○Epileptic females spent 20.6 s [0–49.8].○Non-epileptic females spent 11.1 s [0–26.7].


In summary, there were no significant differences in the primary anxiety indicator measured in EPM between epileptic and non-epileptic participants, regardless of their sex.

Duration of self-grooming, s (an indicator of displacement behavior under stress, Appendix A):
○Epileptic males self-groomed for 57.9 s [34.2–97.6].○Non-epileptic males—for 78.1 s [47.0–126.7].○Epileptic females—for 61.7 s [21.0–94.6].○Non-epileptic females—for 51.5 s [26.2–68.3].


In general, duration of self-grooming behavior in the EPM varied across groups, indicating a lack of significant effect of epilepsy or sex on this displacement behavior in EPM.

The number of head dips (an indicator of exploratory behavior):
○Epileptic males had 8.5 head dips [4.0–13.0].○Non-epileptic males—6.0 head dips [4.0–9.0].○Epileptic females—10.5 head dips [7.0–10.0].○Non-epileptic females—8.5 head dips [6.5–13.5].


In all, exploratory behavior in the EPM did not differ significantly between the groups.

The number of rearings (an indicator of risk assessment behavior):
○Epileptic males displayed 14.0 rearings [11.0–20.0].○Non-epileptic males—15.0 [14.0–20.0].○Epileptic females—15.0 [12.0–17.5].○Non-epileptic females—18.5 [14.0–26.0].


These results indicated that risk assessment behavior was similar across the groups.

Overall, the data showed that absence epilepsy did not significantly affect anxiety, displacement behavior (self-grooming), exploratory, or risk assessment behaviors in the EPM test. However, individual variability within each group should be noted.

Distribution of time in open arms (TOA in *s*) and duration of self-grooming (T*grooming* in *s*) were not normal (Kolmogorov-Smirnov test, *p* > 0.05), suggesting the presence of different anxiety-related behavioral strategies. K-means cluster analysis of TOA and T*grooming* revealed three statistically significant clusters (Figure 3a), which might represent three distinct anxiety-like behavioral phenotypes:Low anxiety in 13 rats (11%) with median TOA = 5.2 s [0–21.4] and median T*grooming* = 46.8 s [25.5–68.5].Self-grooming strategy in 25 rats (21%) with median TOA = 21 s [0–29.5] and median T*grooming* 154 s [126.7–184].High anxiety in 81 rats (68%) with median TOA = 84 s [79–107] and median Tgrooming = 44 s [68–78].

The number of epileptic and non-epileptic subjects, irrespective of sex, did not differ significantly across different anxiety-related clusters (Figure 3b, Fisher exact test, all *p* values in chi-square statistics >0.05). Consequently, neither sex nor epilepsy influenced anxiety-like behavioral phenotypes in WAG/Rij rats.

Composite behavioral indices for the EPM did not significantly differ between epileptic and non-epileptic subjects. However, notable differences were observed in the Anxiety Composite Index (Figure 4a), which assessed open-arm avoidance and physiological stress markers. Specifically, non-epileptic female rats exhibited higher ACI scores, indicative of greater anxiety, compared to epileptic females (Kruskal–Wallis H(1;48) = 5.3, *p* = 0.021). Therefore, non-epileptic females exhibited by a higher level of anxiety associated with markers of physiological stress compared to epileptic females. In contrast, male rats did not show such a difference.

The Generalized Anxiety Index, which incorporated self-grooming behavior, was unaffected by absence epilepsy (Figure 4b). Similarly, the Exploration-Motor Index, which contrasts active exploration (rearing and head dips) with inhibition (grooming), was not influenced by epilepsy (Figure 4c). Most rats had negative EMI scores, suggesting inhibition or stereotypy, while only a few exhibited positive scores indicative of active exploration.

The Exploratory Drive Profile (Figure 4d), measuring the ratio of exploratory to self-directed behaviors, did not exceed 1 in most rats and was not impacted by absence epilepsy. Notably, WAG/Rij rats displayed a very low degree of active exploration in the EPM, irrespective of epilepsy or sex.

The Risk Assessment Ratio (Figure 4e), which evaluated cautious exploration relative to open-arm entries, showed that cautious exploration in the EPM was not altered by absence epilepsy.

Overall, non-epileptic profile in female WAG/Rij rats linked to an increase of generalized anxiety index. However, exploratory behavior, including active exploration, inhibition, and cautious exploration, did not appear to be influenced by absence epilepsy.

### 3.2. Domain-Specific Composite Metrics

The effect of absence epilepsy in the cognitive domain was first examined using results the active avoidance test, a method for assessing associative learning within the context of fear conditioning. This part of the study involved 60 rats, comprising 41 males (23 epileptic and 18 non-epileptic) and 19 females (14 epileptic and five non-epileptic). The test consisted of 50 presentations of the CS and US, and the learning criterion was set at achieving five avoidances within a sequence of six consecutive trials. All female rats successfully met the learning criterion, while 22% of male rats (9 out of 41) refused to perform the task.

We took into account the speed of task acquisition in 51 successful learners (32 males and 19 females) by determining the number of trials required to meet the learning criterion. The distribution of trials to criterion was heterogeneous (Figure 5a), with a remarkable drop in acquisition speed at the 21st trial, indicating a potential learning plateau or difficulty in further acquisition. Therefore, two distinct subgroups were identified: good learners (≤21 trials, n = 22 rats) and bad learners (>21 trials, n = 29 rats). The proportion of good and bad learners did not significantly differ between epileptic and non-epileptic subjects, irrespective of sex (Figure 5b; Fisher exact test, all *p* > 0.05).

Among males, 31% (10/32) of epileptic subjects were classified as good learners, compared to 10% (1/10) of non-epileptic subjects. However, this difference was not statistically significant. The distribution of good learners was non-uniform across sex and epilepsy, limiting reliability of further analysis. Consequently, our focus shifted to the performance of bad learners. Anxiety-related impairments in avoidance learning were assessed using the Anxiety-Latent Avoidance Failure. This metric integrated the scores of generalized anxiety (GAI) from the EPM test with the outcomes of active avoidance test. Among male bad learners, epileptic subjects exhibited significantly higher ALAF scores compared to non-epileptic ones (Kruskal–Wallis H(1;19) = 5.06, *p* = 0.025; Figure 5c). Higher ALAF scores were associated with anxiety-related impairments in avoidance learning.

These findings suggest that while task acquisition speed does not differ significantly between epileptic and non-epileptic learners, yet epileptic males may experience greater anxiety-related impairments in avoidance learning tasks.

The other domain-specific composite metrics introduced in Section 2.3.6 (Cognition, Anxiety, Motor, and Exploration) were not significantly affected by absence epilepsy or by sex (Kruskal–Wallis tests, all *p*’s > 0.05,). More specifically,

**Cognition**. The median score for epileptic males was 29.0 [24.3–32.0], indicating a relatively high level of cognitive flexibility. The scores for non-epileptic males (27.3 [19.3–30.8]), epileptic females (35.0 [20.7–38.5]), and non-epileptic females (27.1 [26.5–33.2]) showed variability, but generally aligned with the high score indicating intact cognitive flexibility.**Anxiety**. The lowest median score was observed in non-epileptic males (0.69 [0.27–1.29]), followed by non-epileptic females (1.02 [0.37–2.55]), epileptic females (1.05 [1.00–2.86]), and epileptic males (1.47 [0.34–4.38]). The higher score of composite Anxiety metric for epileptic subjects suggested slightly higher anxiety levels, but overall, anxiety levels were low across all groups.**Motor**. The high median score for non-epileptic males (1.258 [0.845–1.322]) suggested strong motor capabilities and confidence. Epileptic males had a median score of 0.925 [0.700–1.318], epileptic females—0.894 [0.739–1.343]), and non-epileptic females—1.154 [0.772–1.324]. These scores indicate generally good motor performance across all groups, suggesting that absence epilepsy does not significantly impair motor function.**Exploration**. The scores for all groups were positive, indicating a tendency to explore, although the ranges varied. The median scores for epileptic males was 0.425 [−0.012–0.647]), for non-epileptic males—0.477 [0.229–0.606]), epileptic females—0.486 [0.035–0.834] and non-epileptic females—0.561 [0.416–0.800]. These scores suggested a generally consistent level of exploration tendencies across all groups.

Overall, absence epilepsy did not seem to have a significant impact on cognitive flexibility, anxiety levels, motor capabilities, or exploration tendencies within the framework of our study. Additionally, sex did not significantly affect these metrics.

### 3.3. Cross-Domain Composite Metrics

The composite metric of Social-Emotional Competence was derived from the combined results of the Social Recognition, Social Preference, and Social Dominance tests. Lower scores on this metric indicated heightened vulnerability to social stress. A statistically significant effect of absence epilepsy was observed only in subjects with high levels of anxiety (Figure 6a), but not in those with anxiety-associated grooming behaviors or low anxiety levels. Among subjects with high anxiety, epilepsy was associated with higher scores on the Social-Emotional Competence metric (122.7 [84.3–163.6]) compared to non-epileptic subjects (67.7 [27.6–129.5]), as assessed by the Kruskal–Wallis test (H(1;70) = 7.01, *p* = 0.0078). Therefore, absence epilepsy in WAG/Rij rats, regardless of sex, was linked to increased vulnerability to social stress.

The Anxiety-Avoidance Axis metric was constructed by integrating data from the EPM, sucrose preference, and active avoidance tests. It specifically evaluated behavioral inhibition associated with heightened anxiety in the EPM, anhedonia, and anxiety-induced reductions in exploratory behavior during the active avoidance test. Higher scores on this metric were indicative of a passive coping strategy. Again, a statistically significant effect of absence epilepsy was only evident in subjects with high anxiety levels (Figure 6b), with no significant differences observed in other groups. Among subjects with high anxiety, epileptic rats exhibited higher scores on the Anxiety-Avoidance Axis (5.6 [2.8–7.5]) compared to non-epileptic rats (2.8, [1.9–4.1]), as determined by the Kruskal–Wallis test (H(1;39) = 4.69, *p* = 0.030). These results indicate that absence epilepsy in WAG/Rij rats enhances passive coping strategies or behavioral inhibition.

The Exploratory-Learning Style metric was derived from the EPM and active avoidance tests, assessing adaptive learning based on risk-assessment exploration and rapid avoidance learning. Notably, absence epilepsy did not impact this metric, suggesting that the learning style of epileptic and non-epileptic WAG/Rij rats did not differ.

## 4. Discussion

Our primary goal was to explore the neurobehavioral comorbidities associated with absence epilepsy in WAG/Rij rat model, with a specific focus on anxiety and potential sex-related differences. All rat subjects were diagnosed with absence epilepsy using a new, non-invasive scalp EEG technique [32]. This method was both efficient and safe, enabling continuous monitoring of the disease and preventing injuries that are crucial for behavioral research. Our subjects were genetically predisposed to absence epilepsy, yet only some exhibited EEG signs of the disease, such as sustained SWDs, while others remained non-epileptic until at least 13 months of age. This finding parallels the human condition, where individuals with a genetic predisposition to absence epilepsy may not develop the disease. Absence epilepsy has a genetic basis, with 16–45% of cases having a positive family history. However, disease penetrance is incomplete, with concordance rates of 70–85% in monozygotic twins and 33% in first-degree relatives (cited by [33,34,35]). Over several years, we observed variability in the severity of absence epilepsy in the Moscow population of WAG/Rij rats [36,37,38,39,40], making this model more reflective of human childhood absence epilepsy (CAE).

Here we analyzed direct anxiety measures in the EPM test for anxiety. We found no significant differences in anxiety (time in open arms), stress-related displacement behavior (self-grooming), or exploratory behavior (head dips, rearings) between epileptic and non-epileptic participants, regardless of sex.

We also evaluated composite behavioral indices from the EPM test, which were designed to provide a more holistic view on anxiety, exploratory behavior, and stress responses. The following composite behavioral indices were unaffected by absence epilepsy: GAI, which incorporated time in open arms, physiological stress markers and grooming; EDP, which assessed exploratory behaviors vs. self-directed behaviors, and RAR, which assessed exploration relative to open-arm entries. However, absence epilepsy did affect the Anxiety Composite Index. This index combined time spend in open-arms with physiological stress markers, such as defecation and urination frequency. Non-epileptic females exhibited higher ACI scores than epileptic females, suggesting a greater anxiety and associated physiological stress in non-epileptic females, compared to epileptic females. Male rats did not show such a difference.

The question arises: Why was only the ACI composite measure significantly affected by absence epilepsy, while GAI, EDP, and RAR were not? The ACI could be affected by absence epilepsy because it incorporates a specific aspect, such as acute autonomic nervous system reactivity under stress. This reactivity is directly influenced by the neurophysiological changes associated with epilepsy and exhibits significant sexual dimorphism. In contrast, GAI, EDP, and RAR, which evaluate more complex or relative behavioral responses, were likely affected by the inherent behavioral profile of the WAG/Rij strain (low exploration, high inhibition) or were insensitive to the specific dysregulation of autonomic nervous system functions caused by epilepsy. This emphasizes the critical importance of using direct physiological measures and conducting sex-specific data analyses when studying anxiety-related comorbidities in epilepsy models.

In the current study, we categorized rats based on their behavior in the EPM test for anxiety using K-means clustering and defined three anxiety-like behavioral phenotypes: low anxiety, self-grooming strategy and high anxiety. The majority of rats (81%) exhibited high anxiety, 21% displayed prolonged self-grooming behavior with a grooming time almost three times higher than other groups, and only 11% showed low anxiety. The proportion of epileptic and non-epileptic rats within each anxiety-like behavioral phenotype did not significantly differ, regardless of their sex. It is intriguing that a substantial number of rats (21%) displayed extended self-grooming in the EPM test. This test creates an approach-avoidance conflict by presenting novelty and the fear of open spaces. Rats in the “grooming cluster” likely experienced an unresolved conflict, leading them to engage in displacement grooming as a redirected behavior to cope with stress. Displacement behaviors are most pronounced in ambiguous or threatening environments, such as the open arms of the EPM, where neither full exploration nor complete avoidance occurs. Our data suggest that self-grooming in the EPM was not linked to extreme anxiety (81% of rats were classified as high-anxiety) or low anxiety (11%). Instead, it may reflect an intermediate stress response. This response involves arousal that is high enough to inhibit exploration but not severe enough to trigger freezing or escape behaviors [41,42].

When examining the impact of absence epilepsy on cognitive functions, we assessed the effectiveness of learning. Specifically, we measured whether rats could achieve the learning criterion, which required them to successfully avoid five CS out of six consecutive trials. We noted the tendency for sex-related differences in learning efficacy: 100% of females met the learning criterion, but only 78% of males performed this task. This is in line with our earlier results indicating that female WAG/Rij rats performed active avoidance task better than males [36]. Here we categorized subjects into two subgroups based on their performance in an active avoidance task: good learners and bad learners. Our analysis revealed that there was no significant difference in task acquisition speed between epileptic and non-epileptic learners. However, among male poor learners, epileptic subjects had a higher ALAF score compared to non-epileptic. The ALAF score combined the generalized anxiety level in the EPM test and the outcomes of active avoidance test. This difference between ALAF scores suggested that epileptic male poor learners might experience more anxiety-related difficulties in avoidance learning tasks compared to their non-epileptic counterparts.

In the domain-specific cognitive composite metric, integrating social memory (social recognition test), associative memory (active avoidance within a fear conditioning paradigm), and cognitive flexibility, we found no significant effect of absence epilepsy status or sex. This contrasts with numerous studies reporting cognitive impairments in WAG/Rij rats, predominantly in males, across various specific tests (e.g., [28,43,44,45]). The absence of expected effect on cognitive composite metric may be attributable to the chronic nature of epilepsy in the WAG/Rij rat model. We tested rats at the age of 8–13 months, when the absence epilepsy phenotype has been established, but cognitive deficits might not yet be fully manifested or could be partially compensated for. Our testing window may have missed critical periods where specific cognitive deficits could be detected using our tasks. Additionally, most studies on WAG/Rij rats have focused predominantly on males to avoid confounding effects of the estrous cycle, resulting in a paucity of data on cognitive deficits in females. This scarcity of female data limits our ability to draw reliable conclusions regarding sex differences in cognitive function within this strain. To mitigate potential hormonal influences, we conducted an associative memory test in female rats during their diestrus state. Future studies in both male and female rats are needed to better understand epilepsy-associated cognitive deficits and any sex-specific effects. This approach will enhance the construct validity of WAG/Rij rat model and deepen our understanding of cognitive deficits associated with absence epilepsy.

Anxiety comorbidity is a complex and context-dependent phenomenon. Clinical studies have shown that people with epilepsy often have anxiety disorders [2,3]. However, our findings are consistent with studies in GAERS and NEC (summarized in Table 1), which suggest that anxiety-like behaviors are not always a direct or uniform consequence of seizures ([23,24] see Table 1 for references). We did not define the significant effect of absence epilepsy in WAG/Rij rats on the GAI (incorporating self-grooming, a displacement behavior linked to anxiety) and the RAR (reflecting cautious exploration), therefore, core aspects of anxiety-like behavior in the EPM are independent from the absence epilepsy phenotype at the studied age (8–13 months). These results are consistent with the notions that comorbid anxiety in absence epilepsy involves distinct neural circuits beyond the SWD-producing thalamocortical neuronal network (see references in [46]). Therefore, although the amygdala and prefrontal cortex are implicated in both absence seizures and anxiety [47], disruption in the anxiety-related neural pathways in WAG/Rij rat model may be not sufficient to affect the measures in the EPM test.

The profound and consistent negative EMI scores and EDP scores below 1 across all WAG/Rij rats, irrespective of epilepsy status, is striking. This strongly suggests that reduced exploratory drive and increased inhibition/stereotypy represent a core behavioral trait inherent to the WAG/Rij genetic background, rather than a direct consequence of the epileptic phenotype per se.

In our study, we did not find clear differences between epileptic and non-epileptic rats in specific behavioral domains, such as cognition, anxiety, motor function, and exploration. This lack of significant effects could be attributed to several factors. First, there may be variability in epilepsy severity within the epileptic group, which could lead to diverse behavioral responses among the rats. Second, over time, rats may develop compensatory strategies to adapt to their condition. These strategies could potentially mask the effects of epilepsy on their behavior. Finally, some rats may develop coping mechanisms to maintain normal behavior and adapt to experiencing absence seizures.

Our analysis revealed that the effects of absence epilepsy on complex behavioral metrics in WAG/Rij rats depended upon their baseline anxiety profiles: high anxiety, anxiety-associated grooming, and low anxiety. In rats with high anxiety, absence epilepsy was associated with increased scores in social-emotional competence. These scores measured socio-affective functions by combining results from social recognition, social preference, and social dominance tests. Consequently, absence epilepsy enhanced socio-affective responses in high-anxiety rats, potentially due to heightened sensitivity to social stimuli or altered social coping mechanisms. In these high-anxiety rats, absence epilepsy also led to elevated scores on the anxiety-avoidance axis. This metric quantified passive coping strategies by integrating anxiety levels in the EPM, anhedonia (measured via sucrose preference), and reduced exploration during active avoidance tasks. Thus, absence epilepsy exacerbated passive avoidance behaviors in high-anxiety rats. Interestingly, absence epilepsy did not significantly impact exploratory learning styles across all anxiety clusters. This finding indicates that core adaptive learning processes remain intact regardless of the rats’ anxiety state.

In general, behavioral comorbidities of absence epilepsy in the WAG/Rij rat model are not uniform but are critically modulated by the individual’s inherent anxiety phenotype. Epilepsy specifically potentiates socio-affective reactivity (increased social-emotional competence) and passive coping strategies (increased anxiety-avoidance axis) only in individuals exhibiting a high-anxiety baseline. This highlights anxiety state as a pivotal moderator of behavioral comorbidity of absence epilepsy in WAG/Rij rat model, suggesting shared or interacting neural substrates for high anxiety and susceptibility to specific epilepsy-related behavioral alterations.

### Potential Clinical Applications

Our findings, while derived from a rat model (WAG/Rij), offer several hypotheses and potential clinical applications for clinical practice in human settings. Below, we attempt to translate our key findings into concepts relevant to human diagnosis, treatment, and patient management.

**Finding 1.** Absence epilepsy in females may reduce autonomic stress reactivity (i.e., defecation and urination), suggesting a chronic neuroendocrine adaptation with unchanged observable anxiety-like behavior.

**Clinical application.** Clinicians should be aware that standard psychiatric interviews or questionnaires, which focus on thoughts and behaviors, might underestimate the physiological component of anxiety in female patients with absence epilepsy. These patients may report feeling anxious but exhibit fewer physical signs, such as palpitations, sweating, and gastrointestinal distress.

Assessment could be enhanced by incorporating objective measures of autonomic function, such as heart rate variability, skin conductance response, and cortisol levels, during stress-provoking tasks or interviews. This would provide a more comprehensive understanding of their anxiety profile.

This blunted physiological response could be a compensatory mechanism. Thus, prescribing standard anti-anxiety medications, such as SSRIs, should be done cautiously, as the underlying neuroendocrine system is already altered. Non-pharmacological interventions, such as cognitive behavioral therapy and mindfulness, which target the cognitive aspects of anxiety, might be particularly beneficial.

Understanding this blunted physiological response helps explain the complex relationship between epilepsy and anxiety. It suggests that the relationship is not simply a matter of “more anxiety” or “less anxiety”, but rather a qualitative change in how anxiety manifests physiologically, particularly in females.

**Finding 2.** Absence epilepsy did not cause a general cognitive deficit in fear conditioning setup, but females showed a significant advantage in active avoidance learning. In contrast to females, absence epilepsy in males exacerbated anxiety-related learning difficulties.

**Clinical application.** Educators and neuropsychologists should be aware that learning difficulties in children with absence epilepsy, particularly boys, are often driven by high anxiety in learning contexts rather than by pure cognitive deficits. Implementing strategies to reduce test anxiety and create low-stress learning environments can significantly improve academic outcomes. Neuropsychological assessments should carefully distinguish scores on tasks sensitive to anxiety (e.g., timed tests or tasks with negative feedback) from scores measuring pure cognitive capacity. This distinction is crucial for accurate interpretation of test results.

The finding underscores the importance of sex-specific approaches in clinical management. Boys with absence epilepsy may need additional support to address anxiety-mediated learning barriers, while girls can potentially leverage their cognitive strengths. For boys experiencing academic challenges, interventions should focus on managing anxiety through therapy or anxiety management techniques, in addition to providing educational support.

**Finding 3.** The behavioral impact of absence epilepsy varies significantly among subjects, largely due to their pre-existing anxiety levels. In subjects with high anxiety, absence epilepsy can enhance social-emotional skills but also lead to more passive coping strategies.

**Clinical application.** This finding underscores the importance of routinely screening for anxiety in all epilepsy patients at diagnosis. The clinical needs and presentation of a high-anxiety epilepsy patient will differ markedly from those of a low-anxiety patient. Early identification of high-anxiety individuals can help predict those who are likely to develop passive coping strategies, such as avoidance or helplessness, which are risk factors for depression. By flagging these patients early, psychological interventions can be implemented to build active coping skills. Moreover, recognizing heightened social sensitivity in high-anxiety epilepsy patients can be beneficial. While it may manifest as empathy, it can also make individuals more vulnerable to social rejection. Therapeutic approaches could focus on leveraging this sensitivity as a strength while addressing its challenges.

The shared circuitry hypothesis suggests that epilepsy and neurobiological comorbidities, including anxiety. overlap in neural networks, including the amygdala, prefrontal cortex, and thalamocortical circuits [2,48]. Treatments targeting these shared pathways, such as neuromodulation or specific medications, could potentially be more effective by addressing both conditions simultaneously.

**Finding 4.** All WAG/Rij rats (epileptic and non-epileptic) showed high behavioral inhibition and low exploration, suggesting a strong genetic influence on anxiety-like behavior, which is analogous to an anxious temperament in humans.

**Clinical application.** In cases of genetic epilepsy, healthcare providers can help families understand that a inherited anxious temperament might be a separate, co-occurring trait in the family lineage, not just a consequence of the seizures. This can reduce the stigma associated with anxiety and position it as part of an individual’s neurobiological profile.

For pediatric patients with a family history of both epilepsy and anxiety, healthcare professionals can monitor for early signs of behavioral inhibition, such as timidity or aversion to new experiences. Early behavioral interventions can be implemented to build resilience and coping skills before clinical anxiety develops.

## 5. Conclusions

This study investigated neurobehavioral comorbidities in the WAG/Rij rat model of absence epilepsy, focusing on anxiety, cognitive function, and sex differences. Key findings include:Standard behavioral indices in the EPM (time in open arms, grooming, exploration) were unaffected by absence epilepsy, suggesting that core anxiety-like behaviors in this test are independent of the epileptic phenotype at the studied age (10–11 months). However, the Anxiety Composite Index, which incorporated physiological stress markers (defecation, urination), revealed a sex-specific effect: non-epileptic females exhibited higher ACI scores than epileptic females, indicating greater autonomic stress reactivity. This suggests that absence epilepsy may blunt physiological anxiety responses in females, possibly due to chronic neuroendocrine adaptations.Cognitive composite metrics showed no epilepsy- or sex-related effects, contrasting with previous reports. Females outperformed males in active avoidance learning (100% vs. 78%). Epilepsy exacerbated anxiety-related learning difficulties in males.The impact of absence epilepsy was modulated by baseline anxiety. In high-anxiety rats, epilepsy increased social-emotional competence (potentially reflecting heightened social sensitivity) and passive coping strategies (elevated anxiety-avoidance axis scores). This highlights that anxiety-like behavioral phenotype is a critical moderator of epilepsy-related behavioral changes, pointing to shared or interacting neural circuits.All WAG/Rij rats showed low exploration and high behavioral inhibition, as evidenced by negative EMI and EDP <1 across all rats, correspondingly. This may suggest a strong genetic influence on behavior, independent of epilepsy.

In summary, our study highlights the complex interplay between absence epilepsy, anxiety, and cognition, emphasizing the importance of baseline anxiety and genetic background in epilepsy research.

## Figures and Tables

**Figure 1 biomedicines-13-02075-f001:**
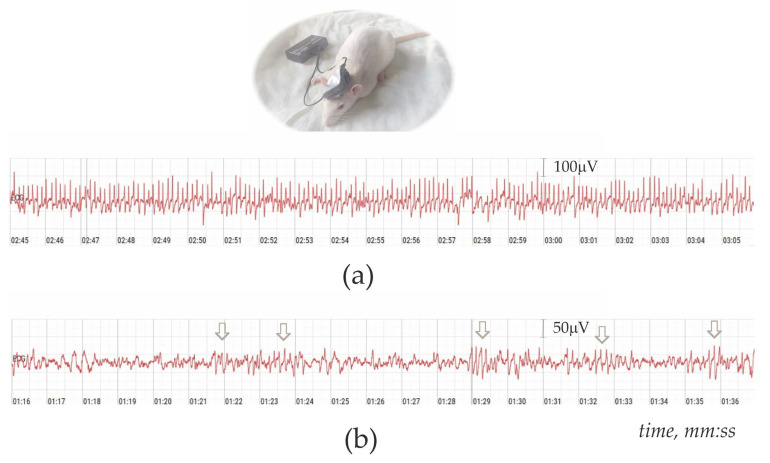
Non-invasiveEEG-based diagnostic of absence epilepsy in rats after intraperitoneal injection of xylazine (2 mg/kg). (**a**) EEG record with continuous and stereotypic 8 Hz spike-wave discharges (SWDs) in a male subject with absence epilepsy. (**b**) EEG record with spindle-like ~6 Hz spike-wave bursts in non-epileptic female subject. Arrows indicate brief spike-wave complexes.

**Figure 2 biomedicines-13-02075-f002:**
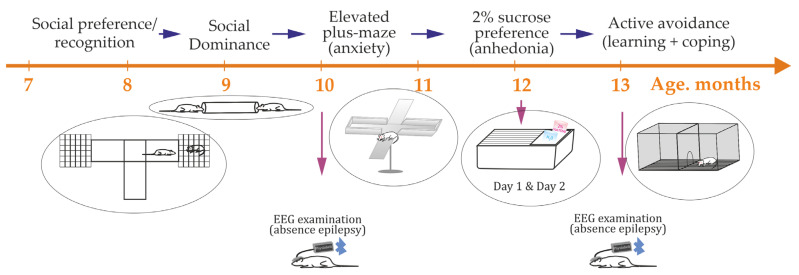
The timeline of behavioral phenotyping: battery of tests.

**Figure 3 biomedicines-13-02075-f003:**
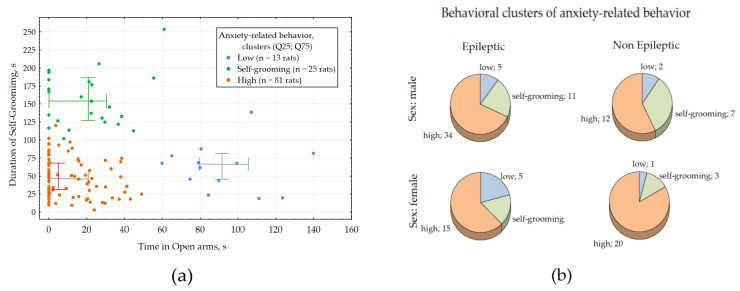
Results of the EPM test for anxiety in WAG/Rij rats. (**a**) Graphic representation of the major anxiety-related behavioral strategies defined with the K-means cluster analysis of time spent in open arms and self-grooming. (**b**) Pie plot showing distribution of different anxiety-related strategies in epileptic and no-epileptic subjects of both sexes.

**Figure 4 biomedicines-13-02075-f004:**
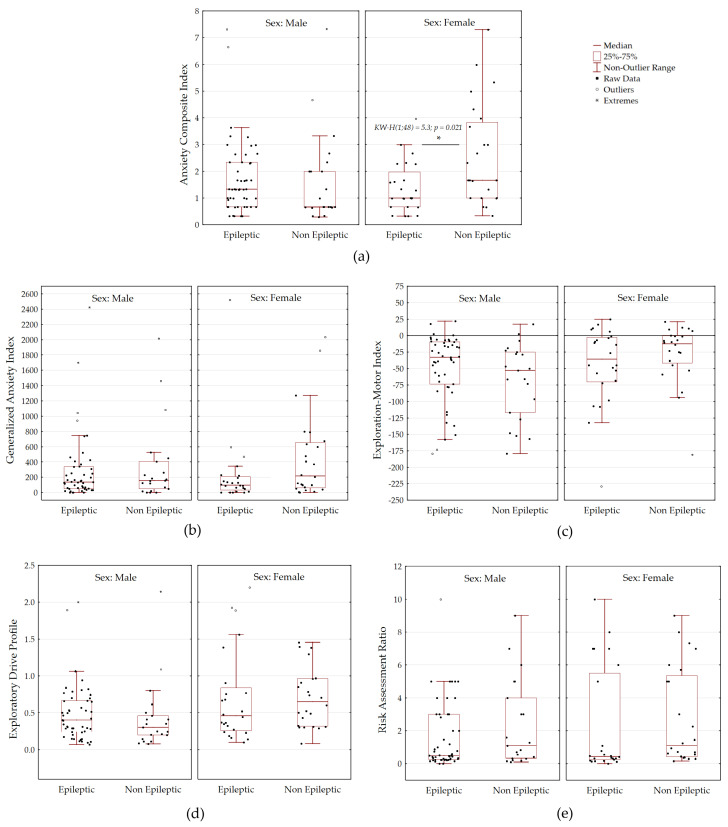
Composite behavioral indices characterizing multidimensional behavioral profile in epileptic and non-epileptic WAG/Rij rats in the EPM (n = 119). (**a**) Anxiety Composite Index, (**b**) Generalized Anxiety Index, (**c**) Exploration-Motor Index, (**d**) Exploratory Drive Profile, (**e**) Risk Assessment Ratio *—significant differences according to Kruskal–Wallis test, *p* < 0.05.

**Figure 5 biomedicines-13-02075-f005:**
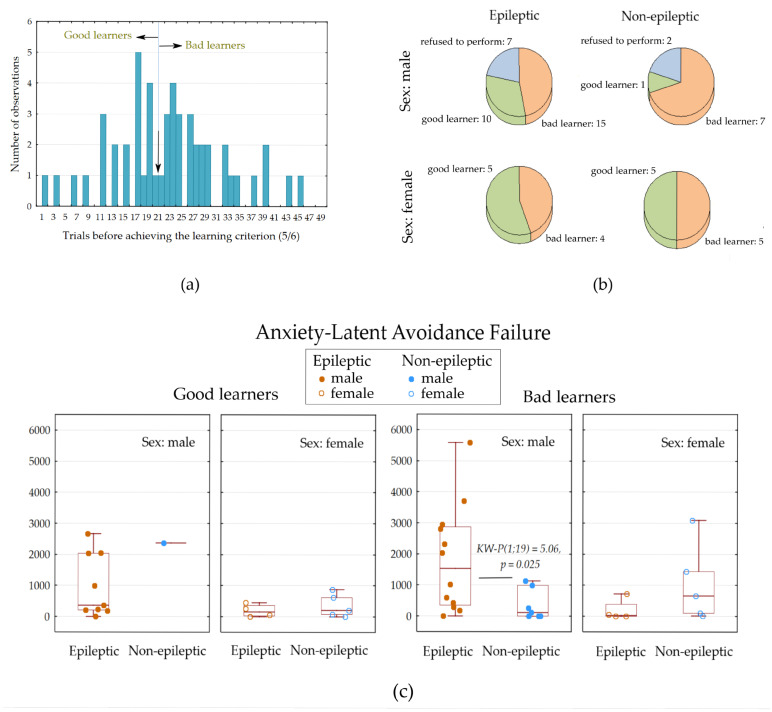
Results of active avoidance fear conditioning test in WAG/Rij rats: direct measures in cognitive domain. (**a**) Distribution of acquisition speed: The histogram shows the number of trials required for rats to achieve the learning criterion (five consecutive avoidances within a sequence of six trials) with a remarkable drop at the 21st trial. (**b**) Classification of learners: A pie chart categorizes the rats into three groups: good learners (≤21 trials), bad learners (>21 trials), and subjects who refused to perform the task. (**c**) Anxiety-Latent Avoidance Failure (ALAF) scores that integrated the generalized anxiety index, GAI, in the EPM test with the outcomes of active avoidance test. Higher ALAF scores indicated anxiety-related impairments in avoidance learning.

**Figure 6 biomedicines-13-02075-f006:**
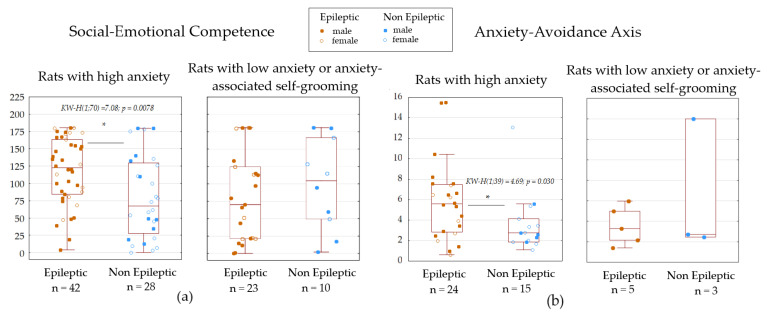
Cross-domain composite metrics characterizing multidimensional behavioral profile in epileptic and non-epileptic WAG/Rij rats. (**a**) Social-Emotional Competence, (**b**) Anxiety-Avoidance Axis. *—significant differences according to Kruskal–Wallis test, *p* < 0.05.

**Table 1 biomedicines-13-02075-t001:** Outcomes of the EPM testing and behavioral indices in rats with absence epilepsy.

Strain	Sex	Age (Months)	Open Arm Time (%)	Open Arm Entries	Additional Behavioral Indices	Methodological Notes	Ref.
GAERS vs. NEC rats	Males, females	1.6 and 3	Lower % than NEC	No mention	- OFT: Reduced center time (thigmotaxis), fewer rearings. - Defecation: Higher boli count (stress indicator). - Locomotor Activity: Reduced total distance traveled.	Tested during light phase (inactive period); 5-min EPM session.	Jones et al., 2008 [22]
GAERS vs. NEC rats	Males, females	1.2–2.1	Lower % than NEC	Fewer than NEC	- ASR: Elevated startle amplitude. - Light-Dark Box: Reduced transitions. - Plasma Corticosterone: Elevated post-test.	Combined EPM and ASR testing; SWD frequency inversely correlated with open arm time.	Marks et al., 2016 [11]
Long-Evans vs. Wistar	Males	9–12	Higher % than Wistar	Higher than Wistar	- OFT: No locomotor differences. - SWD Correlation: Negative association between SWD count and open arm time. - Marble Burying: Reduced in Long-Evans (lower anxiety).	Nocturnal testing (active phase); SWD monitored via EEG.	Shaw et al., 2009 [27]
WAG/Rij vs. Wistar	Males	5 and 13	No difference	Higher than Wistar	- Closed Arm Entries: Increased in WAG/Rij. - Home Cage Activity: Hyperactivity in novel environments. - Social Interaction: No strain difference.	Testing during light phase; 10-min EPM session.	Karson et al., 2012 [28]
WAG/Rij vs. Wistar	Males	3–4 and 5–6	No difference	No mention	- Audiogenic Seizures: Higher anxiety in susceptible rats. - FST: Increased immobility (depressive-like behavior).	Audiogenic priming used; EPM conducted after audiogenic seizure induction	Sarkisova & Kulikov, 2006 [26]
GAERS vs. NEC rats	Males, females	1.6 and 3	Lower % than NEC	No mention	- OFT: Reduced center time (thigmotaxis), fewer rearings. - Defecation: Higher boli count (stress indicator). - Locomotor Activity: Reduced total distance traveled.	Tested during light phase (inactive period); 5-min EPM session.	Jones et al., 2008 [22]
GAERS vs. NEC rats	Males, females	1.2–2.1	Lower % than NEC	Fewer than NEC	- ASR: Elevated startle amplitude. - Light-Dark Box: Reduced transitions. - Plasma Corticosterone: Elevated post-test.	Combined EPM and ASR testing; SWD frequency inversely correlated with open arm time.	Marks et al., 2016 [11]
Long-Evans vs. Wistar	Males	9–12	Higher % than Wistar	Higher than Wistar	- OFT: No locomotor differences. - SWD Correlation: Negative association between SWD count and open arm time. - Marble Burying: Reduced in Long-Evans (lower anxiety).	Nocturnal testing (active phase); SWD monitored via EEG.	Shaw et al., 2009 [27]
WAG/Rij vs. Wistar	Males	5 and 13	No difference	Higher than Wistar	- Closed Arm Entries: Increased in WAG/Rij. - Home Cage Activity: Hyperactivity in novel environments. - Social Interaction: No strain difference.	Testing during light phase; 10-min EPM session.	Karson et al., 2012 [28]
WAG/Rij vs. Wistar	Males	3–4 and 5–6	No difference	No mention	- Audiogenic Seizures: Higher anxiety in susceptible rats. - FST: Increased immobility (depressive-like behavior).	Audiogenic priming used; EPM conducted after audiogenic seizure induction	Sarkisova & Kulikov, 2006 [26]
GAERS vs. NEC rats	Males, females	1.6 and 3	Lower % than NEC	No mention	- OFT: Reduced center time (thigmotaxis), fewer rearings. - Defecation: Higher boli count (stress indicator). - Locomotor Activity: Reduced total distance traveled.	Tested during light phase (inactive period); 5-min EPM session.	Jones et al., 2008 [22]

Abbreviations: acoustic startle response (ASR), electroencephalography (EEG), forced swim test (FST), open field test (OFT), spike-wave discharges (SWD).

## Data Availability

All experimental data obtained in the current study are shown in figures and tables. Primary datasets are available from the corresponding author upon reasonable request.

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
