# Peer review of "Behavioral Phenotyping of WAG/Rij Rat Model of Absence Epilepsy: The Link to Anxiety and Sex Factors"

_biomedicines, 2025, doi:10.3390/biomedicines13092075_

Round 1
Reviewer 1 Report
Comments and Suggestions for Authors
- The title and research purpose should be consistent. The title emphasizes anxiety, but the statement of research purpose does not mention anxiety.
- The mention of 'prioritizing anxiety as a key modulator' in the abstract seems to indicate that anxiety is an upstream event and cause of other behavioral phenotypes. Suggest citing references in the introduction or discussion section to explain this viewpoint.
- The abstract mentions' safety phenotypes', it is suggested to explain this name in the methods section.
- Regarding the number of animal cases, the initial count was 131, followed by 119 for testing purposes, with the missing 12 requiring explanation.
- The method section should include explanations of grouping and the number of animals in each group.
- The annotations of Figures 4 and 5 should indicate the n values of each group.
Author Response
Thank you for taking the time to review our paper and for providing valuable feedback. We appreciate your effort and have carefully addressed all of your comments. We have highlighted the revised text in the resubmitted docx file.
1. Questions for General Evaluation
Are the methods adequately described? Must be improved. The Method section has been revised for clarity and completeness.
Are all figures and tables clear and well-presented? Must be improved. We have added the missing information to Figures 5 and 6.
2. Point-by-point response to Comments and Suggestions for Authors
Comment 1. The title and research purpose should be consistent. The title emphasizes anxiety, but the statement of research purpose does not mention anxiety.
Response. We agree with the reviewer primary aim was anxiety-related neurobehavioral comorbidities. We corrected the Abstract accordingly: "This study aims to investigate anxiety-related neurobehavioral comorbidities and potential sex differences in aged WAG/Rij rats, a well-established animal model of absence epilepsy.
Comment 2. The mention of 'prioritizing anxiety as a key modulator' in the abstract seems to indicate that anxiety is an upstream event and cause of other behavioral phenotypes. Suggest citing references in the introduction or discussion section to explain this viewpoint.
Response. We changed 'a key modulator' by 'a key influencing factor'. Thank you for pointing out that anxiety often serves as a primary and upstream event in epilepsy. We explained this in Introduction: "Anxiety often serves as a primary and upstream event in epilepsy, causing and shaping other neurobehavioral comorbidities, such as depression and cognitive deficits ..." and provided some references.
Comment 3. The abstract mentions' safety phenotypes', it is suggested to explain this name in the methods section.
Response. The reviewer means 'anxiety phenotypes'. The term 'anxiety phenotypes' is not precise and was replaced with 'anxiety-like behavioral phenotypes'. The "-like" suffix signals that rats' behavior is a model for the human condition.
The reviewer's comment suggests that the description of 'anxiety phenotypes' should be included in the Methods section. However, the anxiety-like behavioral phenotypes were actually defined based on the results of the Elevated Plus Maze (EPM) test. These phenotypes — low anxiety, self-grooming strategy, and high anxiety — were identified through the analysis of the test results. The definition of these phenotypes is given in the Results and Discussion sections.
Comment 4. Regarding the number of animal cases, the initial count was 131, followed by 119 for testing purposes, with the missing 12 requiring explanation.
Response. Thank you for noting this discrepancy. The initial count was indeed 131. However, data from only 119 rats was available for the EPM test to assess anxiety. The remaining 12 rats were excluded because they either fell off the platform or their video recordings were corrupted. We have provided a detailed explanation for these cases in the manuscript and corrected the text accordingly.
Comment 5. The method section should include explanations of grouping and the number of animals in each group.
Response. We have clarified the grouping and the number of animals in the Methods section. Due to limitations in room capacity and test equipment, not all rats could participate in every test. The number of rats that successfully passed each test and whose data were analyzed is indicated in the appropriate Results section, and we have added this information to Figures 5 and 6.
Comment 6. The annotations of Figures 4 and 5 should indicate the n values of each group.
Response. Ok.
Reviewer 2 Report
Comments and Suggestions for Authors
General comment:
This is a great review article and providing timely information to clinicians.
The authors emphasized that absence epilepsy is a common pediatric neurological disorder characterized by brief seizures and lapses in awareness. The relationship between anxiety and absence epilepsy is complex. Therefore, they conducted a comprehensive behavioral phenotyping study in aged WAG/Rij rats to investigate neurobehavioral comorbidities and sex differences. Specifically, they tested 131 WAG/Rij rats using a sequence of behavioral tests, including anxiety (elevated plus maze), anhedonia (sucrose preference), social function, and associative learning (fear conditioning). Multidimensional metrics assessed cognition, motor function, and exploration strategies, prioritizing anxiety as a key modulator. They noticed that EEG phenotyping identified epileptic and non-epileptic rats. In the elevated plus maze, traditional anxiety measures showed no significant differences, but the Anxiety Composite Index revealed higher autonomic reactivity in non-epileptic females. Cognitive assessments showed no epilepsy- or sex-related differences in overall learning, but females outperformed males in avoidance learning. Epileptic males with poor learning had higher anxiety-avoidance scores. High-anxiety rats exhibited enhanced socio-affective reactivity and passive coping, with no effect on exploratory learning. Therefore, they achieved their conclusion that comorbidities are not uniform but modulated by inherent anxiety phenotypes. More precise phenotyping of the WAG/Rij model can improve its translational value in understanding epilepsy-associated psychiatric disorders.
Generally speaking, the current study was well-written and their methodology sounds scientifically. I read with interest. I believed that this manuscript will bring important scientific contributions to the world.
Minor comments:
- In page 1 line 15, please spell out EEG when it first appeared.
- As the authors addressed, there had been several animal models regarding absence epilepsy, the authors could address the reason why they choose WAG/Rij rat model in their study but not other rat models.
- As clinicians might have interest, the authors could add further statements regarding the application of their findings in clinical practice in human settings.
- Since this is an animal study, has it followed any guidelines, such as ARRIVE? If so, please provide related information in detail.
- Has this study been registered in any open registry database before they started their study?
Author Response
Dear reviewer, thank you very much for taking the time to carefully examine our work and provide positive feedback. We truly appreciate your efforts and attention to detail. All of your comments have been addressed, and the changes in the text are highlighted in red.
Comment 1. In page 1 line 15, please spell out EEG when it first appeared.
Response. Thank you for noting this issue. EEG has been spelled out in Abstract and where it first appeared.
Comment 2. As the authors addressed, there had been several animal models regarding absence epilepsy, the authors could address the reason why they choose WAG/Rij rat model in their study but not other rat models.
Response. This point is crucial to our work because we routinely use the WAG/Rij rat model of absence epilepsy in our basic research. The translational significance of the WAG/Rij model has been explained in the Introduction section: "...Our research focuses on the WAG/Rij rat model, which exhibits an exceptional degree of face validity, construct validity, and predictive validity for absence epilepsy.."
Comment 3. As clinicians might have interest, the authors could add further statements regarding the application of their findings in clinical practice in human settings.
Response. Thank you for your insightful comment. We agree that the potential clinical applications of our findings are of significant interest. In response, we have added Section 4.1, "Potential clinical applications," to Discussion. This section includes four key points that highlight the practical implications of our results for clinical practice. We believe this addition will help to better communicate the translational value of our work.
Comment 4. Since this is an animal study, has it followed any guidelines, such as ARRIVE? If so, please provide related information in detail.
Response. The experiments were conducted in accordance with Directive 2010/63/EU of the European Parliament and of the Council on the Protection of Animals used for scientific purposes and approved by the animal ethics committee of our institute (protocols No. 4 dated October 26, 2021 and additional protocol No. 4 dated December 13, 2022). We used the ARRIVE guidelines for reporting animal research, completed the ARRIVE checklist and sent it to Assistant Editor.
Comment 5. Has this study been registered in any open registry database before they started their study?
Response. No, this study was not registered in any open registry database before they started their study. Prospective registration was not a standard practice for this type of study (an exploratory research).
Round 2
Reviewer 2 Report
Comments and Suggestions for Authors
The authors had addressed all my comments. I had not further comment to be addressed.